# Magnitude and associated factors of peripheral cytopenia among HIV-infected children attending at University of Gondar Specialized Referral Hospital, Northwest Ethiopia

**Biruk Bayleyegn** *, **Berhanu Woldu, Aregawi Yalew , Fikir Asrie**

Department of Hematology and Immunohematology, College of Medicine and Health Science, University of Gondar, Gondar, Ethiopia

* birukbayle@gmail.com

## Abstract

### Background

Isolated or multi lineage cytopenia are the most common clinicopathological features and independently associated with increased risk of disease progression and death among human immunodeficiency virus infected children. In the study area, there is scarcity of data about the magnitude of various cytopenia.

### Objectives

Aimed to determine the magnitude and associated factors of peripheral cytopenia among HIV infected children at the University of Gondar Specialized Referral Hospital ART clinic, Northwest Ethiopia.

### Methods

Institutional based cross-sectional study was conducted on 255 HIV infected children from January- April 2020. None probable convenient sampling technique was used to select the study participant. Socio demographic data were collected by pre tested structured question-naire via face-to-face interview and their medical data were obtained from their follow-up medical records. Moreover, blood specimens were collected and examined for complete blood count, viral load and blood film, whereas stool specimens were collected and examined for intestinal parasites. Bi-variable and multi-variable logistic regression models were fitted to identify associated factors of cytopenia. P-Value <0.05 was considered as statistically significant.

### Result

The overall magnitude of peripheral cytopenia was 38.9%. Anemia, leukopenia, lymphopenia, thrombocytopenia and bi-cytopenia were 21.2%, 12.2%, 11%, 1.6% and 3.9%

**Data Availability Statement:** All relevant data are within the manuscript.

**Funding:** The authors received no specific funding for this work.

**Competing interests:** The authors have declared that no competing interests exist.

**Abbreviations:** AIDS, Acquired Immunodeficiency Syndrome; BAZ, Body Mass Index for Age Z score; BM, Bone Marrow; EDTA, Ethylene Di amine tetra acetic Acid; HAART, Highly Active Antiretroviral Treatment; HAZ, Height-for-Age Z score; Hgb, Hemoglobin; HIV, Human Immunodeficiency Virus; IQR, Inter Quartile Range; MCH, Mean Cell Hemoglobin; MCHC, Mean Cell Hemoglobin Concentration; MCV, Mean Cell Volume; MPV, Mean Platelet Volume; OI, Opportunistic Infection; RBCs, Red Blood Cells; RDW, Red Cell Distribution Width; UoGSRH, University of Gondar Specialized Referral Hospital; WBCs, White Blood Cells; WHO, World Health Organization.

respectively. Being in the age group of 2–10 years (AOR = 5.38, 95%CI 2.33–12.46), AZT based regimen (AOR = 5.44, 95%CI: 2.24–13.21), no eating green vegetables (AOR = 2.49, 95% CI: 1.26–4.92) and having plasma viral load >1000 copies /ml (AOR = 5.38, 95%CI: 2.22–13.03) showed significant association with anemia.

## Conclusion

Anemia was the predominant peripheral cytopenia among HIV infected children in this study. It was strongly associated with AZT based drug type, age below 10 years and high viral load. Critical stress should be given for early investigation and management of cytopenia in addition to the use of alternative drug which leads to higher viral suppression and lower risk of toxicity issue.

## Introduction

Hematological abnormalities are the most common clinicopathological features of human immunodeficiency virus (HIV) infection. Of these hematological abnormalities, peripheral cytopenia is become the most common manifestation, which is a reduction of any of the blood cell lines leading to leukopenia, anemia, and thrombocytopenia in the peripheral blood among patients with HIV infection [1]. In both antiretroviral-treated and untreated individuals, cytopenia is independently associated with an increased risk of disease progression and death [2]. The pathophysiological mechanisms responsible for HIV related cytopenia are quantitative and qualitative marrow defects and immune cytopenia. So far, the reduction of colony forming unit of granulocyte, erythrocyte, monocyte and megakaryocyte (CFU-GEMM) and erythropoietin production by HIV itself and indirectly through HIV proteins such as the envelope protein and abnormal levels of cytokine in the bone marrow (BM), prolonged use of anti-retroviral therapy (ART), opportunistic infection (OI) and malignancies are a key mechanism for HIV associated cytopenia [3–6]. The incidence and severity of cytopenia in HIV-infected children seem to be dependent on the level of viral replication. Since children are potentially at a higher risk of developing viremia due to weight-based dosing, poor tolerability of drugs and suboptimal adherence [4, 7, 8].

Anemia is the most common peripheral cytopenia in people living with HIV. It affects 12.3% to 90% of HIV-infected children and was a strong associated factor of morbidity and mortality [9–12]. Although HIV associated anemia is multifactorial, the principal factors are infiltration of the BM by neoplasm or infection, myelo-suppressive medications such as Zidovidine and HIV infection by decreasing the production of endogenous erythropoietin. Moreover, red blood cells (RBC) auto-antibodies and the use of various medications may result the consequence of hemolytic anemia. The severity of anemia also increases with advanced stage of HIV disease, the presence of intestinal parasites, malaria, OI and high level of viremia [13–15]. Thrombocytopenia is the second most common hematological abnormality found in children with HIV infection. This frequent disorder occurring in about 30–40% of individuals with HIV infection. Thrombocytopenia may occur in all stages of HIV infection and it was identified as a poor prognostic factor for progression of HIV infection to acquired immunodeficiency syndrome (AIDS) and death. But it may increase in associated with high level of viremia in the plasma, type of highly active anti-retroviral therapy (HAART), effects of OI, malignancies and an alteration of the immune system [16–18]. Leukopenia occurs in about

one-third of children with untreated HIV infection predominantly due to lymphopenia and neutropenia. Reduction in absolute number of CD4+ T-cell lymphocytes occur as one of the earliest immunologic abnormalities of HIV infection and an important prognostic indicator of risk of developing OI. Like anemia and thrombocytopenia, leukopenia has also been reported as due to adverse effects of HAART [19–21]. Generally, peripheral cytopenia such as anemia, leukopenia, and thrombocytopenia in associated with HIV infection and treatment can impair the quality of life of people living with HIV/AIDS, particularly children. For this reason the hemoglobin (Hgb) concentration, white blood cell count (WBC), total lymphocyte count and other hematological parameters have been proposed as alternative markers of the disease, especially for developing countries where financial resources are limited [16, 22, 23]. A number of studies have been conducted on cytopenia mainly anemia and among HIV-infected adults. However, there is a scarcity of data about the magnitude of various peripheral cytopenia of all major blood cell lineage among HIV infected children and there is no enough information to associate with viral load. Therefore, this study was carried out to investigate the magnitude and associated factors of peripheral cytopenia among HIV infected children attending at University of Gondar specialized referral hospital, Northwest Ethiopia.

## Materials and methods

### Study setting and population

Institutional based cross-sectional study was conducted at University of Gondar Specialized Referral Hospital (UOGSRH) ART clinic from January to April 2020. A total of 255 HIV positive children (2 HAART naive and 253 HAART experienced) who attended the UOGSRH ART clinic during the study period were included consecutively. After getting written consent and assent, study participants were enrolled in to the study if they were aged between 6 months to 15 years old. Meanwhile, HIV positive children who were severely sick due to other medical conditions, such as having confirmed chronic renal failure and liver disease, on radiation therapy and immunosuppressive chemotherapy in the previous 45 days, were excluded from the study due to the fact that these may unambiguously affect the hematological values. In addition, those children on treatment for anemia, neutropenia or thrombocytopenia as well as who had traumatic injury or surgical interventions resulting in blood loss and without a legal guardian or unaccompanied children during the study period were not included in this study.

The sample size was computed using single population proportion formula by considering the following assumptions: from cytopenia the prevalence of anemia was 30.9%, far greater than those with thrombocytopenia and leukopenia [24] with 5% margin of error and 95% confidence level then yielding a sample size of 328. Since the source population was <10,000, finite population correction formula is applied by taking an estimated population of 536 HIV infected children at ART clinic during the study period, and then the final sample size was 203. However, to increase the power of the study and since the available data was more than required, 255 available samples were considered for analysis and were included in this study. None probable convenient sampling technique was applied to recruit study participants who meet the inclusion criteria.

### Data collection and laboratory procedures

**Socio-demographic and clinical data collection.** Socio-demographic characteristics of children (such as age, gender, residence, educational status) and socioeconomic/demographic characteristic of family/caregivers, including (age, gender, family income, family size, occupation, education and life status) were collected using a pre-tested structured questionnaire via a face-to-face interview technique. Moreover, detailed clinical data of the children, such as

world health organization (WHO) HIV disease stage, OI, type of HAART and HAART experience and duration of HAART were collected by reviewing the medical record of HIV infected children. Anthropometric measurements (height and weight) were taken using calibrated equipment's by a trained clinical nurse working at pediatric ART clinic and *Z*-scores of nutritional indices, such as weight-for-age (WAZ), height-for-age (HAZ), weight for height (WHZ) and BMI-for-age (BMA) were scored using WHO Anthro (for children aged ≤5 years) and Anthro-plus (for children aged >5 years) soft ware's.

*Hematological parameters*. After collecting 5ml of venous blood sample in EDTA tube from each study participants, CBC results which include RBC parameters (RBC count, Hgb, mean cell hemoglobin (MCH), mean cell hemoglobin concentration (MCHC) and mean cell volume (MCV)), WBC parameters (WBC count, absolute neutrophil count, neutrophil percentage, lymphocyte count, lymphocyte percentage, mixed count that encompasses eosinophil, basophil and monocyte and mixed cell percentage) and platelet parameter (platelet count and mean platelet volume) were determined using sysmex KX21 hematology analyzer. This was accomplished with in one hours of blood collection by experienced hematologist at University of Gondar referral hospital hematology laboratory.

*HIV RNA detection and quantification*. HIV viral load was determined directly by advanced molecular technique on blood plasm collected from each study participants using plasma separator gel containing EDTA tubes. TAQMAN® AMPLICOR HIV-1 MONITOR (Roche Molecular Systems) which is an in vitro diagnostic nucleic acid amplification test were for the quantification of HIV-1 RNA in human plasma. Template RNA, primer, dNTPmix, were combined with nuclease-free water in a PCR tube. RNase inhibitor and reverse transcriptase was added to the PCR tube. A master mix (containing buffer, dNTP mix, MgCl2, Taq polymerase and nuclease-free water) was added to each PCR tube. Then GAG specific to HIV1 primer was added. PCR tubes were placed in a thermal cycler for 30 cycles of the amplification program, which included three steps: denaturation, annealing and elongation giving real time viral load in copies/μl. Then, the amount of circulating HIV was measured and reported as HIV RNA copies per milliliter (copies/ml) of plasma by well-trained laboratory technologists.

*Parasitological examination*. For intestinal parasite examination, 3-5gm fresh stool was collected by standard clean, wide mouthed, leak proof and dry plastic stool cup from each study participant. The stool specimens were transported in screw-capped cups in 10% formalin from a sample reception area to the laboratory. Test tub flotation concentration procedures were done for the detection of protozoan cysts, helminthic ova and larvae in addition to using the direct wet mount. This was based on the principle that flotation solution must have a higher specific gravity than the parasite egg (parasite eggs are 1.05–1.24, while flotation solutions specific gravity > 1.24). Thereby during examination of the solution in the topmost layer will clearly indicate the presence of eggs, but the fecal material and fiber settle at the bottom. Moreover, both thin and thick blood films using 10% Giemsa were examined for the detection and identification of malaria parasites and any quantitative as well as qualitative cell abnormality were reported by experienced hematologist.

## Operational definition

Peripheral cytopenia:—is defined as reduction in either of the cellular elements of blood, i.e., RBCs, WBCs or platelets [25]. Unicytopenia: defined as patients with isolated anemia, thrombocytopenia or neutropenia. Bi-cytopenia: Defined as having two of the three lineage cell counts below the levels [26]. Anemia:—was defined as Hgb concentration of less than 11 g/dl for 6–59 months of age, less than 11.5 g/dl for ages 5–11 years, and less than 12 g/dl for ages 12–14 years old children. Mild anemia was defined as follows: Hgb 10.0–10.9 g/dl for children

aged 6–59 months, 11.0–11.4 g/dl for children aged 5–11 years and 11.5–11.9 g/dl for children aged 12–15 years. Moderate anemia was defined as Hgb 7.0–9.9 g/dl for children aged 6–59 months and 8–10.9 g/dl for children aged 6–15 years. Severe anemia was also defined as Hgb<7.0 g/dl for children aged 6–59 months and <8.0 g/dl for those aged 6–15 years after altitude adjustment by subtracting 0.8 g/dl since the study area altitude is 2133 as recommended by WHO [27] Anemia is usually classified based on the size of RBCs, as measured by MCV. It can be microcytic if MCV typically less than 80 fl, normocytic 80 to 100 fl, or macrocytic greater than 100 fL. Furthermore, it was defined as hypochromic if MCH was <27 pg [28]. Thrombocytopenia:—Defined as when a platelet count $< 150 \times 10^3/mm^3$, categorized as mild (50,000–149,999 platelets/µl), moderate (20,000–49,999) and severe $< 20,000$ [17] trombocytopenia. Leukopenia:—defined as WBC count <4000/mm$^3$, lymphopenia: as lymphocyte count <1500/mm$^3$ and neutropenia: NC of <1500/mm$^3$, severity further classified as follows: 750–1500/mm$^3$mild neutropenia, 500–750/mm$^3$moderate neutropenia, <500/mm$^3$severe neutropenia and <250/mm$^3$severe life-threatening neutropenia [24, 29].

## Data quality assurance

To assure the quality of data, training was given for data collectors prior to data collection and daily close supervision was made during the data collection period. Before the actual data collection date, the questionnaire was pre-tested on 5% of the sample size at Gondar poly health center ART clinic. Then, necessary modification was done based on its analysis. Blood sample was checked whether they were in the acceptable criteria like; free of hemolysis, no clotting, sufficient volume, correct labeling and collection time. All samples were analyzed in one laboratory (University of Gondar specialized referral hospital Hematology laboratory section) with the same hematology analyzer and the same trained hematologists. The performance of Sysmex KX-21 hematology analyzer was checked by daily initialization background check and by running three levels of hematology controls (Normal, Low and High) by applying the principles of Westgard rules and Levy Jennings chart to control the automated instruments. On the other hand, COBAS® TaqMan® Negative control, HIV-1 Low Positive control and HIV-1 High Positive Control in each test batch were used to evaluate the performance of the COBAS AmpliPrep/COBAS TaqMan HIV-1 test. Moreover, known malaria-positive and negative blood were used to check the quality and performance of the Giemsa staining reagent, as well as stool examination were performed after checked the specific gravity of the salt solution was above 1.25 using Hydrometer once a week. Weight and height measuring equipment's were calibrated by using known weight and height before starting the data collection. Two readings were obtained for each measurement, and the mean of two anthropometric measurements were calculated and used for the analysis.

## Data analysis and interpretation

Data were entered into EPI-info 4.4 and have been checked and cleaned for completeness and consistency. Then the data were transferred to statistical package for social science (SPSS) version 20 for analysis. Descriptive statics like frequencies, percentages and medians with interquartile ranges (IQR) for skewed statistical distributions were used to summarize the data. To assess the distribution of data, Shapiro-Wilk test was done. Bi-variable and multi-variable logistic regression were used to measure the association of cytopenia with risk factors. The Hosmer-Lemeshow goodness-of-fit test was used to assess the fitness of the model. Variables with a p value less than 0.2 in the bi-variable analysis were fitted into the multi-variable binary logistic regression analysis to control the possible effect of confounding. Both crude odds ratio (COR) and adjusted odds ratio (AOR) with the corresponding 95% confidence interval (CI)

were calculated to measure the strength of association. Finally, in the multi-variable analysis, variables with p values less than 0.05 was considered as statistically significant.

### Ethical consideration

Ethical approval was obtained from Research and Ethics Review Committee of the School of Biomedical and Laboratory Sciences, College of Medicine and Health Sciences, University of Gondar. Furthermore, support and permission letter were secured from UoGSRH. In addition, following an explanation of the purpose, the benefits and the possible risks of the study, written informed consent was taken from a parent/legal guardian and assent was sought from children before commencement of the study. It was made clear that participation in the study were purely on a voluntarily basis and refusal was possible. To ensure confidentiality of data, study participants were coded by using unique codes, and only authorized persons were accessing the collected data. The study participant's with abnormal findings were linked to the physicians who are working at the ART clinic for proper patient care.

## Results

### Socio-demographic and clinical characteristics of study participants

A total of 255 HIV positive children (253 HAART experienced and 2 HAART naïve) were recruited into this study, of which half of them were female 128 (50.2%). The median age of the participants was 13 years with interquartile rang (IQR) (10,14) and majority of the children 189(74.1%) were age between 11–15 years. Based on WHO clinical stage criteria of HIV, most of the study participants 245 (96.1%) were in WHO stage I. About 221 (86.7%) of study participants were under ART treatment for the duration of ≥1 year. According to HAART type, about 106 (41.6%) study participants were taking TDF containing HAART regimen. Nearly half (44%) of the study participants were stunted based on WHO nutritional assessment guidelines. In the current study, only 29 (11.4%) of study participants had history of OI, the commonest being TB infection seen in 16 (6.3%), and pneumonia was seen in 12 (4.7%) followed by fungal infection 1 (0.4). Moreover, from a total 255 HIV infected children who participated in the study, 57 (22.4%) of the children had one or more of the following intestinal parasites (*E.hsitoltica*, *Ascarice lumbricode*, *Tenia spp*, *Hook Worm*, *H.nana*, *Ascaris and E.histolitica*, *Ascaris and H.nana*, *trophozoit of Gardia lambilia* and *oocyst of cryptosporidium* spp). Consequently, in this study 80 (31.4%) of the study participants had viral load ≥1000 copies/ml (> 3log10) (Table 1).

### Magnitude of cytopenia among study participants

The overall magnitude of peripheral cytopenia were 38.8%. The magnitude of isolated anemia, leukopenia, lymphopenia, thrombocytopenia and bi-cytopenia among HIV infected children were 21.2%, 12.2%, 11%, 1.6% and 3.9% respectively (Table 2). Among the bicytopenia cases, almost all had anemia and leukopenia together. Based on severity, from a total of anemic children: 4(7.4%), 21(38.9%) and 29(53.7%) had sever, moderate, and mild anemia, respectively while none of the participants had sever thrombocytopenia and leukopenia.

Hemoglobin value and platelet count were showed negatively correlation with viral load (r = -0.02 and -0.03) respectively. However, a study didn't find a significant correlation between hematological parameters and viral load (p>0.05) (Table 3).

The type of anemia also assessed in this study. Thus, among the total number of cases, the most common type of anemia was normocytic normochromic anemia (74%) (Fig 1).

**Table 1. Socio-Demographic and clinical characteristics of HIV infected children and their family/ guardian who visited UoGSRH ART clinic, Gondar, Northwest Ethiopia, 2020 (N = 255).**

| Socio-Demographic characteristics | | | Clinical characteristics | | |
|---|---|---|---|---|---|
| Variables | Category | Frequency (%) | Variables | Category | Frequency (%) |
| **Gender of children** | Male | 127(49.8) | **WHO staging** | I | 245(96.1) |
| | Female | 128(50.2) | | II | 10(3.9) |
| **Age (years) children** | ≤5 | 9(3.5) | **Viral load** | Not detected | 109(42.7) |
| | 6–10 | 57(22.4) | | ≤1000copies /ml | 66(25.9) |
| | 11–15 | 189(74.1) | | ≥1000 copies/ml | 80(31.4) |
| **Residence** | Urban | 226(88.6) | **BAZ** | Wasted | 45(17.6) |
| | Rural | 29(11.4) | | Normal | 210(82.4) |
| **Educational status of children** | No formal education | 22(8.6) | **HAZ** | Stunted | 111(43.5) |
| | Primary school | 201(78.8) | | Normal | 144(56.5) |
| | Secondary school | 32(12.5) | | | |
| **Family income per month** | ≤ 1000 | 123(48.2) | **OIs** | Yes | 29(11.4) |
| | 1001–2000 | 78(30.6) | | No | 226(88.6) |
| | >2000 | 54(21.2) | | | |
| **Family size.** | ≤4 | 183(71.8) | **Presence of diarrhea** | Yes | 20(8.8) |
| | 4–6 | 56(22) | | No | 235(92.2) |
| | >6 | 16(16) | | | |
| **Marital status of caregiver** | Married | 139(54.3 | **HAART experience** | Naïve | 2(0.8) |
| | Single | 7(2.7) | | ≤6months | 20(7.8) |
| | Separated | 23(9) | | 6-12month | 12(4.7) |
| | Widowed | 86(33.7) | | >12month | 221(86.7) |
| **Parental status** | Both live | 143(56.1) | | | |
| | Father live | 18(7.1) | | | |
| | Mother live | 62(24.3) | | | |
| | Both dead | 32(12.5) | | | |
| **Relationship to children** | Mother | 160(62.7) | **HAART Types** | AZT Containing | 78(30.6) |
| | Father | 54(21.2) | | TDF Containing | 106(41.6) |
| | Other care giver * | 41(16.1) | | ABC Containing | 71(27.8) |
| **Family occupational status** | Privately employed | 31(12.2) | **Eating animal products** | Yes | 183(71.8) |
| | Government employed | 63(24.7) | | No | 72(28.2) |
| | Merchant | 75(29.4) | | | |
| | House wife | 65(25.5) | | | |
| | Other | 21(8.2) | | | |
| **Family educational status** | No formal education | 86(33.7) | **Eating dark vegetable** | yes | 179(70.2) |
| | Primary school | 84(32.9) | | no | 76(29.8) |
| | Secondary and above | 85(33.3) | | | |
| **HIV status of caregiver** | Positive | 215(84.3) | **Intestinal parasite** | Yes | 57(22.4) |
| | Negative | 26(10.2) | | No | 198(77.6) |
| | Not known | 14(5.5) | | | |

**Note:**—Other care giver * = Female care giver, Male caregiver, Funded organization.

**Abbreviation**; AZT = zidovudine,TDF = Tenofovir, ABC = Abacavir.

Furthermore, the higher frequency (41.2%) of anemia was detected among children with high viral load (>1000copies/ml) compared to children with undetected viral load as shown in the (Fig 2) below.

**Table 2. Magnitude of cytopenia in HIV infected children, at UoGSRH, Northwest Ethiopia, 2020 (N = 255).**

| Cytopenia | | Frequency(N) | Percentage (%) |
|---|---|---|---|
| Anemia | Yes | 54 | 21.2 |
| | No | 201 | 78.8 |
| Leukopenia | Yes | 31 | 12.2 |
| | No | 224 | 87.8 |
| Neutropenia | Yes | 1 | 0.4 |
| | No | 254 | 99.6 |
| Lymphopenia | Yes | 28 | 11.0 |
| | No | 227 | 89.0 |
| Thrombocytopenia | Yes | 4 | 1.6 |
| | No | 251 | 98.4 |
| Bi-cytopenia | | 10 | 3.9 |
| Uni-cytopenia | | 89 | 34.9 |
| Total cytopenia | | 99 | 38.8 |

This indicated that highest frequency of cytopenia (anemia, leukopenia, thrombocytopenia and bi-cytopenia) were observed among children with high viremia.

## Cytopenia and associated factors

Among the variables analyzed in bivariate analyses, age, gender of children, family income, family occupational status, no eating dark green leafy vegetable, BAZ, HAART category and duration of HAART use, WHO's HIV/AIDS clinical staging and viral load were associated with anemia at p value 0.2. However, in multi-variable logistic regression analysis, controlling the possible cofounders, age of 2–10 years (AOR = 5.383 95%CI 2.326–12.460), AZT based HAART regimen (AOR = 5.44, 95%CI: 2.24–13.21), not eating dark green leafy vegetable (AOR = 2.49, 95%CI:1.26–4.92) and viral load >1000copies /ml (AOR = 5.38 95%CI: 2.22–13.03) remained significantly associated with anemia among HIV-infected children (Table 4). Furthermore, gender, family income, WHO staging, HAART type, intestinal parasites were associated with leukopenia in bivariate analysis. However, none of these factors showed statistically significant association with leukopenia in multi-variable logistic regression analysis (Table 5).

## Discussion

Isolated or multi lineage cytopenia due to ineffective hemopoiesis or increased peripheral destruction were the common hematological manifestation among patients with HIV infection. In children with HIV, anemia is the commonest hematological disorder. By itself HIV

**Table 3. Correlation of selected hematological profile with viral load rate, UGCSRH, ART clinic (N = 255).**

| | Viral load | | |
|---|---|---|---|
| Hematological value | Mean ±SD | Pearson Correlation = r | Sig |
| Hgb | 12.9± 1.9 | -0.02 | 0.8 |
| RBCx$10^6$l | 4.4± 1.0 | 0.09 | 0.2 |
| WBCx$10^3$l | 6.5±2.3 | 0.03 | 0.7 |
| NEUT | 45.8 ± 14.2 | 0.02 | 0.8 |
| LYM | 40.3 ±13.1 | 0.02 | 0.7 |
| PLTx$10^3$l | 324.3 ±89.3 | -0.03 | 0.6 |

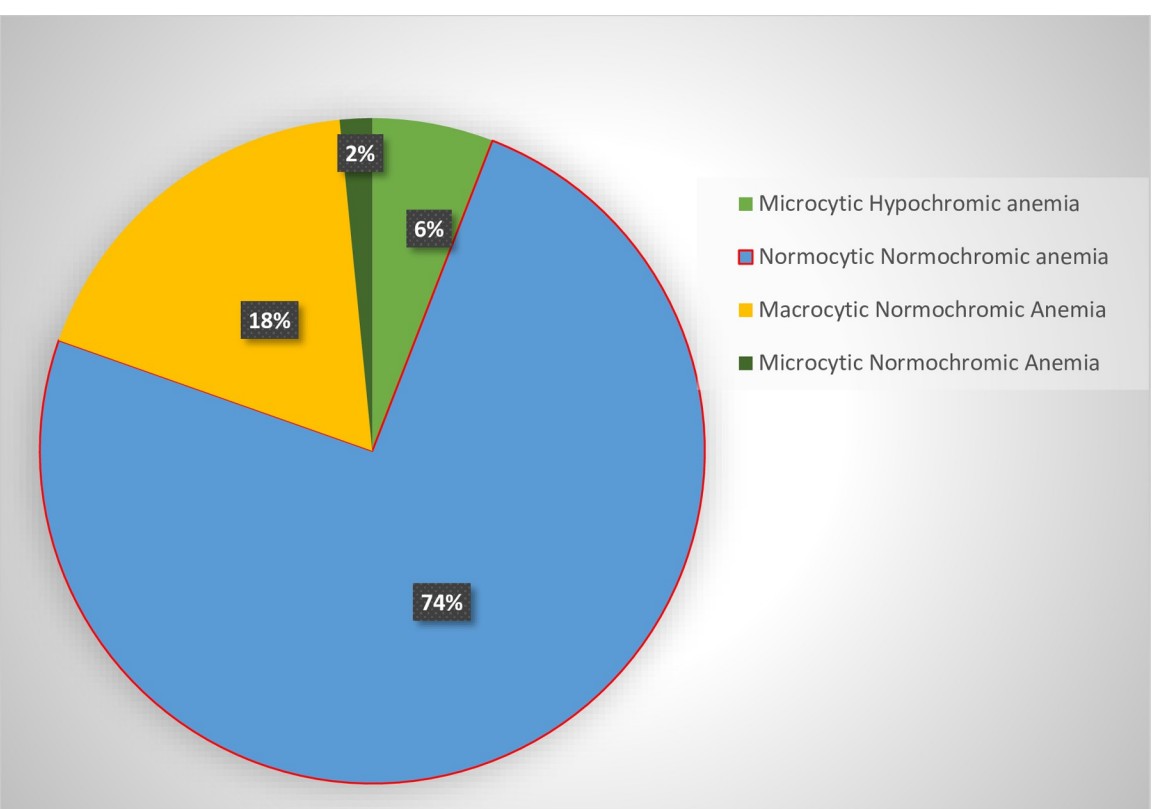

**Fig 1. Type of Anemia detected based on MCV among HIV infected children at UoGCSRH ART Clinic, North West Ethiopia, 2020.**

infection is an obvious cause of most dysfunction, but this is also compounded by opportunistic infections, inflammation, malnutrition and nutritional deficiencies, malignancy and drug toxicity [30].

Anemia and leukopenia especially lymphopenia were common findings in the present study which is in agreement with result documented by different studies [19–21, 31, 32]. The most common hematological derangement in the current study was anemia with over all prevalence of 21.2%. This finding is almost comparable with different literatures conducted by Abebe et al in Jimma, Ethiopia, which is 21.9% [33] and Mihiretie et al in Addis Ababa, Ethiopia 22.2% [34]. However, the prevalence of anemia in this study is low as compared to study done in West Bengal, India, 69% [31], Zimbabwe 30.9% [23], Togo 70.9% [35] and Ghana 63% [32]. These observed differences may be due to differences in age variation of the study participants, sample size difference, diagnostic criteria and cutoff value used to define anemia and epedemicity of blood and intestinal parasite. The most interesting possible reason for the decreased prevalence of anemia in this study could be attributed to the goal and adoption of the recent "WHO 90-90-90 test and treat policy" which brings many more HIV infected children initiated early on HAART thereby reducing anemia comorbidity [36].

Study participants with age of below 10 years had 5 times more risk of developing anemia as compared to those children with age of 11–15 years (95%CI = 2.6–42.6). This is in line with different studies conducted in Ethiopia [24, 37]. This higher prevalence of anemia in children could be explained by the accelerated growth and consequent increased requirement for iron during their development in addition to the disease state and immune destruction [38, 39].

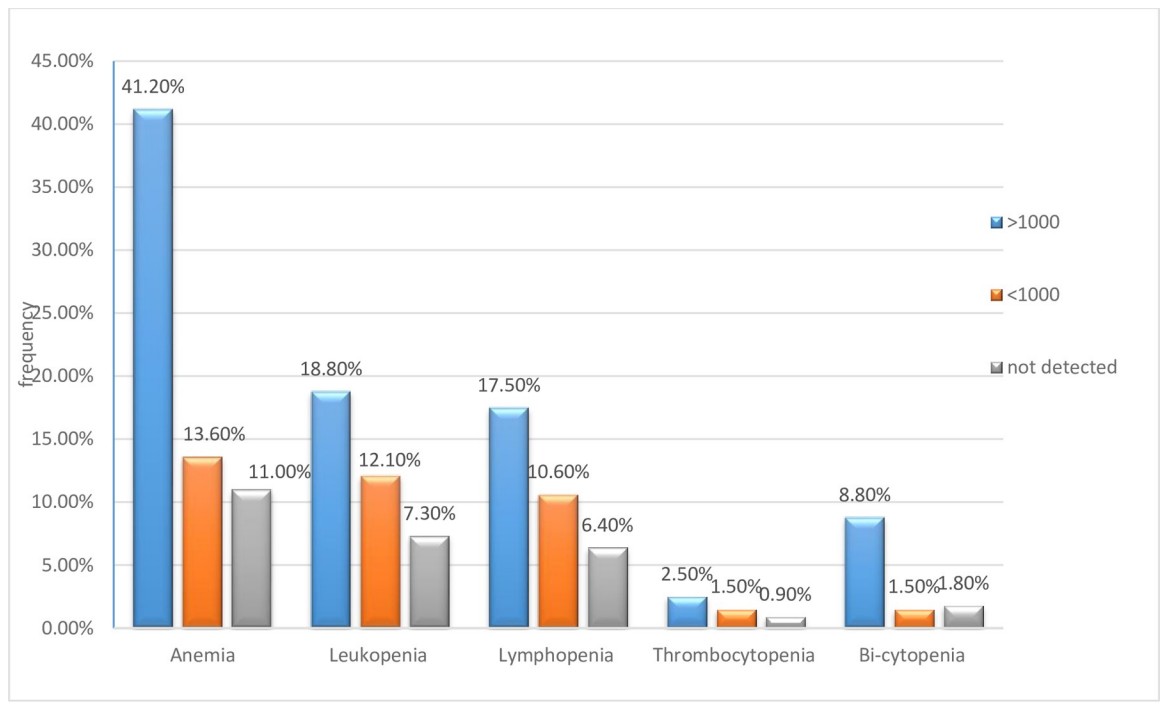

**Fig 2. Comparison of the magnitude of peripheral cytopenia with its viral load rate among study participants at UoGCSRH, North west Ethiopia, 2020.**

However, this result was contradicted with what was reported at Jos, Nigeria, observed that children aged 11–15 years had a significantly lower mean Hgb level as compared to the younger children due to longer duration of HIV infection in this group of children which could have resulted in more severe changes in cytokine production and erythropoietin production [39].

Moreover, this study also revealed that HIV infected children not eating green leafy vegetables were at high risk of developing anemia (AOR = 2.30 95%CI = 1.10–4.82) which is comparable with the study done by Enawgaw et al [40]. This may be due to the fact that dark green leafy vegetables exert the beneficial effects smainly through their high non-heme iron and folic acid content which are contributory cause of anemia [41].

This study also showed that high plasma viral load (viral load > 3log10) was presented in 41.2% of anemic study participants and found to be independently associated with the presence of anemia (AOR = 5.38 95%CI = 2.218–13.047) as compared to their anemic status of the undetected viral load. This could be explained by the fact that an increase in viral activity could be accompanied by a decrease in CFU-GEMM level, reducing production of erythropoietin and increment of apoptosis of erythroid pre-cursors by HIV itself in the children [6]. The result was in agreement with many more literature conducted previously by Ruhinda et al [42], Fenta et al [43], Makubi et al [44] and Shet et al [45]. However, this is not in agreement with a recent study conducted at North Wollo, Ethiopia, which showed no significant difference in the prevalence of anemia [46]. The possible reason encountered to this difference may be due to sample size variation and difference in study design [47].

Furthermore, children on AZT combination HAART regimen was 5 times more risk of developing anemia as compared to those children using other drug regimen (95%CI = 2.2–12.9). The finding in the present study was comparable with the finding reported at Nasarawa

**Table 4. Bi-variable and Multi-variable logistic regression analysis for associated factors of anemia among HIV infected children at UoGSRH, ART Clinic, Gondar, Northwest Ethiopia, 2020 (N = 255).**

| Variable | Category | Anemic (%) | Non-anemic (%) | COR (95% CI) | AOR (95% CI) |
|---|---|---|---|---|---|
| Gender | Male | 32(25.2) | 95(74.8) | 1 | 1 |
| | Female | 22(17.2) | 106(82.8) | 1.62(0.88–2.99) | 0.62(0.34–1.13) |
| Age (years) | 2–10 | 24(36.4) | 42(63.6%) | 3.03(1.60–5.72) | **5.38(2.33–12.46)** *** |
| | 11–15 | 30(15.9) | 159(84.1) | 1 | 1 |
| Residence | Urban | 48(21.2) | 178(78.8) | 1 | 1 |
| | Rural | 6(20.7) | 23(79.3) | 1.03(0.40–2.68) | |
| Family income per month | ≤ 100 | 35(28.5) | 88(71.5) | 0.73(0.25–2.16) | |
| | 1000–2000 | 10(12.8) | 68(87.2) | (1.66–4.95) | |
| | >2000 | 9(16.7) | 45(83.3) | 1 | |
| Family size | ≤ 4 | 42(23.0) | 141(77.0) | 1 | |
| | 4–6 | 11(19.6) | 45(80.4) | 1.17(0.17–2.54) | |
| | ≥6 | 1(6.2) | 15(93.8) | 4.81(0.72–38.16) | |
| Parental status | Both live | 30(21) | 113(79) | 1 | |
| | Father live | 3(16.7) | 15(83.3) | 1.34(0.36–4.89) | |
| | Mother live | 14(22.6) | 48(77.4) | 0.91(0.44–1.87) | |
| | Both dead | 7(21.9) | 25(78.1) | 0.95(0.37–2.4) | |
| Family occupational status | Private | 4(12.9) | 27(87.1) | 1 | 1 |
| | Government | 10(15.9) | 53(84.1) | 0.79(0.23–2.74) | 3.02(0.72–12.70) |
| | Merchant | 15(20) | 60(80) | 0.59(0.18–1.95) | 2.67(0.80–8.98) |
| | Housewife | 19(29.2) | 46(70.8) | 0.36(0.11–1.17) | 1.97(0.62–5.98) |
| | Others | 6(28.6) | 15(71.4) | 0.37(0.09–1.52) | 1.12(0.36–3.48) |
| Family educational status | No | 23(26.7) | 63(73.3) | 0.57(0.17–1.96) | |
| | Primary school | 20(23.8) | 64(76.2) | 0.57(0.19–1.77) | |
| | Secondary and above | 11(12.9) | 74(87.1) | 1 | |
| WHO staging | I | 49(20) | 196(80) | 1 | |
| | II | 5(50) | 5(50) | 0.25(0.07–0.90) | 0.28(0.07–1.10) |
| BAZ | Wasted | 14(31.1) | 31(68.9) | 0.53(0.26–1.08) | 1.90(0.93–3.91) |
| | Normal | 40(19) | 170(81) | 1 | 1 |
| HAZ | Stunted | 27(24.3) | 84(75.7) | 0.73(0.40–1.33) | |
| | Normal | 27(18.8) | 117(81.2) | 1 | |
| OIs | Yes | 8(27.6) | 21(72.4) | 0.67(0.28–1.61) | |
| | No | 46(20.9) | 180(79.1) | 1 | |
| HAART Duration | | | | | |
| | <6months | 4(18.2) | 18(81.8) | 1 | |
| | 6-12month | 5(41.7) | 7(58.3) | 0.31(0.06–1.50) | 0.60(0.18–1.99) |
| | >12month | 45(20.4) | 176(79.6) | 0.87(0.28–2.70) | 3.05(0.84–11.01) |
| HAART Type | AZT Based | 29(38.2) | 47(61.8) | 0.60(0.24–1.52) | **5.32(2.19–12.95)** ** |
| | TDF Based | 17(16) | 89(84) | 0.88(0.44–1.79) | 1.58(0.63–3.94) |
| | ABC Based | 8(11) | 65(89) | 1 | 1 |
| **Eating animal products** | Yes | 35(19.1) | 148(80.9) | 1 | |
| | No | 19(26.4) | 53(73.6) | 0.96(0.47–1.95) | |
| **Eating green leafy vegetable** | Yes | 29(16.2) | 150(83.8) | 1 | |
| | No | 25(32.9) | 51(67.1) | 2.49(1.26–4.92) | **2.30(1.09–4.82)** * |
| Intestinal parasite | Yes | 13(22.8) | 44(77.2) | 0.78(0.31–1.97) | |
| | No | 41(20.7) | 157(79.3) | 1 | |

*(Continued)*

**Table 4.** (Continued)

| Variable | Category | Anemic (%) | Non-anemic (%) | COR (95% CI) | AOR (95% CI) |
|---|---|---|---|---|---|
| Viral load | <1000 copies /ml | 9(13.6) | 57(86.4) | 1.27(0.50–3.20) | 1.27(0.50–3.20) |
| | >1000 copies/ml | 33(41.2) | 47(58.8) | 0.18(0.08–0.37) | **5.38(2.22–13.05)** *** |
| | Not detected | 12(11) | 97(89) | 1 | 1 |

**Note:** * Refers to statistical significance at P<0.05,

** at P value <0.01 and

*** at P value <0.001.

-COR = crud odd ratio, AOR = adjusted odd ratio, CI = Confidence interval.

State, Nigeria [48], Asia [49] and Logas, Nigeria [50] revealing that more anemic patients on AZT combination than on other treatment regimens. This could be due to AZT has been found to exhibit cytotoxicity to the erythroid precursor cells in the bone marrow [30, 51, 52].

In this study from anemic children 4 (7.4%), 21 (38.9%) and 29 (53.7%) of them had sever, moderate, and mild anemia, respectively. The result was consistent with as 4.5%, 39.4% and 56.1% were severe, moderate and mild anemia reported at Bahir-Dar, Ethiopia [37] and 5%,42.5% and 52.2% were severe, moderate and mild anemia respectively done at Addis Ababa, Ethiopia [34]. However, this is lower than a study conducted at Dares Salaam, Tanzania [44], with a severity of 15%. The possible reason attributed to this difference might be due to that majority of the study subjects in Tanzania were in the WHO clinical stage 3 or 4 and high prevalence of hookworm infestation, which means that parasite causes nutritional competition, RBC destruction and impaired nutrient uptake by a direct damage of the intestinal mucosal wall that leads to sever anemia [53]. But in this study, none of participants had late HIV disease stage and no more intestinal parasite was detected due to anti-helmintic drugs given to all HIV infected children as deworming.

As in several other similar studies, normocytic normochromic anemia was the commonest (74%) of anemia, followed by 18% macrocytic normochromic anemia based on MCV classification. This was nearly in line with study at Hawassa, Southern Ethiopia [43], normocytic normochromic anemia was (64.5%) followed by (19.4%) of macrocytic normochromic anemia. This is also comparable with morphological classification reported in Zimbabwe [23] out of 97 children that the most common anemia were normocytic normochromic anemia (61.5%) and Gondar, Ethiopia [40] 46.5% and 39.5% of normocytic-normochromic and macrocytic-normochromic anemia respectively. The occurrence of more percent of normocytic-normochromic anemia on this study might be due to the greater number of study participants in stage 1 disease. However, macrocytic normochromic was probably due to the effect of ART, particularly AZT, which is responsible for the development of macrocytosis (24) since majority of the participants were on HAART.

The second most common peripheral cytopenia in the current study was leukopenia which accounts 12.2% of study population and was not associated with any associated factors. The possible biological explanation of leukopenia in children with HIV could be a decrease in the number of hematopoietic stem cells (HSCs) in the bone marrow, changes in marrow architecture include decreased cellularity and myelodysplasia as well as decreased levels of the factor that stimulates production of white blood cells in the bone marrow (granulocyte colony-stimulating factor (G-CSF)). Furthermore, HIV infection can directly result in lymphopenia as the infection progresses, leading to a decrease in CD4 lymphocytes (31). The prevalence was found to be nearly similar to the findings in the Kenya study [6] where the rate was found to be 10% and Gondar, Ethiopia [24] which reported 12.0%. But in contradiction to this study,

**Table 5. Leukopenia and associated factors among HIV infected children at UoGSRH, ART Clinic, Gondar, Northwest Ethiopia, 2020 (N = 255).**

| Variable | Category | Leukopenia | | COR 95%CI | AOR 95%CI |
|---|---|---|---|---|---|
| | | Yes (%) | No (%) | | |
| Gender | Male | 19(15) | 108(85) | 1 | 1 |
| | Female | 12(9.4) | 116 (90.6) | 0.59(0.27–1.27) | 2.27(0.91–5.66) |
| Age (years) | 2–10 | 9(13.6) | 57 (86.4) | 1.20(0.52–2.75) | |
| | 11–15 | 22(11.6) | 167(88.4) | 1 | |
| Residence | Urban | 29(12.8) | 197(87.2) | 1.99(0.45–8.80) | |
| | Rural | 2(6.9) | 27(93.1) | 1 | |
| Family income per month | ≤ 100 | 15(12.2) | 108(87.8) | 0.69(0.28–1.70) | 1.13(0.33–3.78) |
| | 1000–2000 | 7(9) | 71(91) | 0.49(0.17–1.42) | 1.07(0.30–3.90) |
| | >2000 | 9(16.7) | 45(83.3) | 1 | |
| Family size | ≤ 4 | 20(10.9) | 163(89.1) | 1 | |
| | 4–6 | 8(14.3) | 48(85.7) | 1.36(0.56–3.28) | |
| | ≥6 | 3(18.8) | 13(81.2) | 1.88(0.49–7.17) | |
| Parental status | Both live | 19(13.3) | 124(86.7) | 1 | |
| | Father live | 2(11.1) | 16(88.9) | 0.82(0.17–3.833) | |
| | Mother live | 5(8.1) | 57(91.9) | 0.57(.20–1.61) | |
| | Both dead | 5(15.6) | 27(84.4) | 1.21(.42–3.52) | |
| Family occupational status | Privately | 1(3.2) | 30(96.8) | 0.32(0.03–3.74) | |
| | Government | 11(17.5) | 52(82.5) | 2.01(.41–9.91) | |
| | Merchant | 10(13.3) | 65(86.7) | 1.46(1.46–7.25) | |
| | Housewife | 7(10.8) | 58(89.2) | 1.15(.22–5.99) | |
| | Others | 2(9.5) | 19(90.5) | 1 | |
| Family educational status | No formal education | 11(12.8) | 75(87.3) | 1.10(.44–2.74) | |
| | Primary school | 10(11.9) | 74(88.1) | 1.01(.40–2.58) | |
| | Secondary and above | 4(11.8) | 30(88.2) | 1 | |
| WHO staging | I | 28(11.4) | 217(88.6) | | |
| | II | 3(30) | 7(70) | 3.32(.81–13.59) | 0.58(0.09–3.88) |
| BAZ | Wasted | 4(8.9) | 41(91.1) | 0.66(.22–1.99) | |
| | Normal | 27(12.9) | 183(87.1) | 1 | |
| HAZ | Stunted | 14(12.6) | 97(87.4) | 1.08(.51–2.29) | |
| | Normal | 17(11.8) | 127(88.2) | 1 | |
| OIs | Yes | 5(17.2) | 24(82.8) | 1.60(0.56–4.56 | |
| | No | 26(11.5) | 200(88.5) | 1 | |
| HAART duration | | | | | |
| | <6months | 2(10) | 20(90) | 1 | |
| | 6-12month | 2(16.7) | 10(83.3) | 2.00(.24–16.36) | |
| | >12month | 27(12.2) | 194(87.8) | 1.39(.31–6.29) | |
| HAART Type | AZT Based | 16(21.1) | 62(78.9) | 4.32(1.37–13.64) | 3.35(0.96–11.67) |
| | TDF Based | 11(10.4) | 95(89.6) | 1.94(.59–6.35) | 1.66(0.42–6.58 |
| | ABC Based | 4(5.6) | 67(94.4) | 1 | |
| Eating animal products | Yes | 24(13.1) | 159(89.9) | 0.71(.29–1.74) | |
| | No | 7(9.7) | 65(90.3) | 1 | |
| Eating green leafy vegetable | yes | 22(12.3) | 157(87.7) | 0.96(0.42–2.19) | |
| | No | 9(11.8) | 67(88.2) | 1 | |
| Intestinal parasite | Yes | 11(19.3) | 46(80.7) | 2.13(0.95–4.76) | 2.83(0.08–7.41) |
| | No | 20(10.1) | 178(89.9) | 1 | |
| Viral load | <1000 copies /ml | 8(12.1) | 58(87.9) | 1.67(0.66–4.23) | |
| | >1000 copies/ml | 15(18.8) | 65(81.2) | 0.57(0.21–1.61) | |
| | Not detected | 8(7.3) | 101(93.7) | 1 | |

the prevalence was lower than a study done in West Bengal, India 34% [31] and Zimbabwe 46.4% [23]. The reasons for the observed difference might be due to the heterogeneity of study population (small sample size: 100 in India and 97 in Zimbabwe) and stages of HIV infection (none of study participants were in advanced disease stage in the current study). Moreover, in this study lymphopenia was also assessed and seen in 11% of HIV infected children which was in agreement with a study conducted by Amidu N et.al in Ghana [32] found 11%.

Thrombocytopenia was observed in 1.6% of the cases in the current study. The most common reasons being thrombocytopenia was, peripheral destruction of platelets due to cross-reactivity of HIV antibodies, apoptosis of megakaryocytes due to direct HIV infection and abnormal and dysfunctional platelet production [54]. The prevalence was similar with a finding reported by Oshikoya et al [55] found to be 1.4%. However, this is lower than the prevalence 11.4% reported in DASH Lafia, Nigeria [48] and 11% in India [31]. The lower rates of thrombocytopenia in this study could have resulted from almost all of the study participants were on HARRT and none of advanced disease stage was detected. Hence, advanced HIV infection was decreased platelet production due to direct infection of megakaryocytes in bone marrow [47] but HAART to be associated with increased platelet production, markedly reducing HIV related thrombocytopenia [23]. However, none of these independent factors showed statistically significant association with thrombocytopenia, which is similar to a study reported by Melku et al [37]. This is due to very small number of thrombocytopenic children to predict the association between an exposure and the outcome.

## Limitation of the study

The main limitation of this study is the single site and cross-sectional nature of its design, which makes relationships between cytopenia and associated factors difficult, as it is temporal association. Other limitation of the study is also that the morphological classification of anemia was not determined. In addition, we didn't analyze serum ferritin, vitamin B12 and folate levels and bone marrow examination, which potentially limit this study. Another limitation is that no locally established clinical laboratory reference ranges of healthy and HIV-infected children's in study area, Ethiopia which may lead over or under estimate hematological abnormalities.

## Conclusion

The overall magnitude of peripheral cytopenia was 38.9%. Of which, anemia was the predominant cytopenia among HIV infected children. Children below 10 years, those who had high viral load and not eating green leafy vegetables in addition to close monitoring of children on AZT based regimen were more likely to be anemic. Therefore, critical stress should be given for investigation and management of cytopenia in HIV-infected children, particularly for those who are taking AZT based HAART type, those age below 10 years and high viral load. The use of alternative drug which leads to higher viral suppression and lower risk of toxicity issue need to be encouraged. Moreover, large scale and longitudinal studies are recommended in order to strengthen and explore in depth on the cause of cytopenia.

## Supporting information

**S1 File.**
(DOCX)

## Acknowledgments

First, the authors would like express their deepest gratitude to Department of Hematology and Immunohematology, School of Biomedical and Laboratory Sciences, College of Medicine and Health Sciences, University of Gondar for allowing them to conduct this research. Authors also would like to thank the entire staff of Hematology Laboratory and pediatric ART clinic nurses working at University of Gondar specialized Referral hospital. Finally, authors would like to thank the study participants and their guardians/family for participating in this study.

## Author Contributions

**Conceptualization:** Biruk Bayleyegn.

**Data curation:** Biruk Bayleyegn, Berhanu Woldu, Aregawi Yalew, Fikir Asrie.

**Formal analysis:** Biruk Bayleyegn.

**Funding acquisition:** Biruk Bayleyegn.

**Investigation:** Biruk Bayleyegn.

**Methodology:** Biruk Bayleyegn.

**Project administration:** Biruk Bayleyegn.

**Resources:** Biruk Bayleyegn.

**Software:** Biruk Bayleyegn.

**Supervision:** Berhanu Woldu, Aregawi Yalew, Fikir Asrie.

**Validation:** Biruk Bayleyegn, Berhanu Woldu, Aregawi Yalew, Fikir Asrie.

**Visualization:** Berhanu Woldu, Aregawi Yalew, Fikir Asrie.

**Writing – original draft:** Biruk Bayleyegn.

**Writing – review & editing:** Berhanu Woldu, Aregawi Yalew, Fikir Asrie.

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
