## [Decision Letter · Decision Letter 0]

14 Jan 2021

PONE-D-20-24807

Magnitude and associated factors of peripheral cytopenia among HIV-infected children attending at University of Gondar Specialized Referral Hospital, Northwest Ethiopia.

PLOS ONE

Dear Dr. Bayleyegn,

Thank you for submitting your manuscript to PLOS ONE. After careful consideration, we feel that it has merit but does not fully meet PLOS ONE’s publication criteria as it currently stands. Therefore, we invite you to submit a revised version of the manuscript that addresses the points raised during the review process.

Please ensure you address each of the methodological concerns raised by the reviewers, and provide the requested details to ensure that your study is fully reproducible.

We look forward to receiving your revised manuscript.

Kind regards,

Jamie Males

Senior Editor

PLOS ONE

Journal Requirements:

3. You indicated that you had ethical approval for your study. In your Methods section, please ensure you have also stated whether you obtained consent from parents or guardians of the minors included in the study or whether the research ethics committee or IRB specifically waived the need for their consent.

4. In the Methods section, please provide the specific sequences of the PCR primers used in your study.

"First, the authors would like express their deepest gratitude to Department of Hematology and

Immunohematology, School of Biomedical and Laboratory Sciences, College of Medicine and

Health Sciences, University of Gondar for financial support and allowing them to conduct this

research."

"The authors received no specific funding for this work."

6. We noticed you have some minor occurrence of overlapping text with the following previous publication(s), which needs to be addressed:

https://www.omicsonline.org/scientific-reports/JIDT-SR-662.pdf

https://link.springer.com/chapter/10.1007%2F978-3-030-35433-6_10

http://medcraveonline.com/HTIJ/gender-based-differences-in-hematological-and-cd4-t-lymphocyte-counts-among-hiv-patients-in-ido-ekiti.html

In your revision ensure you cite all your sources (including your own works), and quote or rephrase any duplicated text outside the methods section. Further consideration is dependent on these concerns being addressed.

Reviewers' comments:

Reviewer's Responses to Questions

**Comments to the Author**

1. Is the manuscript technically sound, and do the data support the conclusions?

Reviewer #1: Partly

Reviewer #2: Yes

2. Has the statistical analysis been performed appropriately and rigorously? 

Reviewer #1: No

Reviewer #2: No

3. Have the authors made all data underlying the findings in their manuscript fully available?

Reviewer #1: Yes

Reviewer #2: Yes

4. Is the manuscript presented in an intelligible fashion and written in standard English?

Reviewer #1: No

Reviewer #2: Yes

5. Review Comments to the Author

Reviewer #1: • Thank you for inviting me to review a manuscript entitled with “Magnitude and associated factors of peripheral cytopenia among HIV-infected children attending at University of Gondar Specialized Referral Hospital, Northwest Ethiopia”

• Why none probable convenient sampling technique was used to select the

study participant? b/c the result will not be representative

• The sample size is small since the study was conducted solely in a single institution. In addition why not you didn’t take previous proportions which maximize your sample size. Please justify why?

• Why not you didn’t used a case control study to asses factors?

• Ethical clearance: How blood specimens were collected while most children with HIV are anemic, other hematologic complication and malnutrition? Please send the scan of the IRB permission letter

• How did you get each participants since thy may come at different period of follow up period

• Why you select viral load as a main covariate than other independent variables

• It is not recommended to use the term predictors in cross-sectional study please change it with associated factors throughout the paper.

• You run a logistic regression for only 2 outcome variable (anemia and leukopenia) why not you assessed factors of other components of cytopenia?

• Why not you used other logistic regression model when you have more than 1 outcome variable

• The discussion is not in detail, not including other studies done in Ethiopia among HIV positive children.

Reviewer #2: Comments to authors

I am happy by reviewing this manuscript. For improving the quality of the paper, the following comments were forwarded for the authors.

Title: Magnitude and associated factors of peripheral cytopenia among HIV infected children at the University of Gondar Specialized Hospital ART clinic, Northwest Ethiopia.

1. In the abstract, Background section, the author didn’t show the need/argument in the study. In the result section, the factors which affect cytopenia is not stated rather the author depicts merely for anemia. Besides, the conclusion is not in line with the title.

2. In the Methods and materials part, why the author didn’t include the children less than 6 months. it is a critical question

3. In the result part, please re write the way of putting Interquartile range (IQR) with median

4. How can you operationalize the presence or absence of intestinal parasite for this particular study?

5. Be consistent while you write “multivariable versus multivariate analysis”

6. In the result section, you didn’t touch your title because the author operationalized Peripheral cytopenia as a reduction in either of the cellular elements of blood. The author solely done multivariable model for each parameter independently (anemia, and leucopenia)

7. Besides, the author didn’t address the factors which predicts thrombocytopenia

8. In the discussion part, remove unnecessary sentences like “In this study a total of 255 consecutive HIV infected children were evaluated for peripheral cytopenia on their follow up visit at UOGSRH ART clinic.”

9. Correct all the above comments point by point

6. PLOS authors have the option to publish the peer review history of their article (what does this mean?). If published, this will include your full peer review and any attached files.

Reviewer #1: **Yes: **Biruk Beletew Abate

Reviewer #2: **Yes: **Mohammed Ahmed

---

## [Author Response · Author response to Decision Letter 0]

1 Feb 2021

Response to Reviewers 

Dear, Jamie Males (senior Editor):

Thank you for giving the opportunity to submit a revised version of our manuscript titled Magnitude and associated factors of peripheral cytopenia among HIV-infected children attending at University of Gondar Specialized Referral Hospital, Northwest Ethiopia to journal of Plos One. We grateful to appreciate the time and effort that you and the reviewers have dedicated to providing your valuable feedback on our manuscript. We are grateful to the reviewers for their insightful comments on our paper. We have been able to incorporate changes to reflect most of the suggestions provided by the reviewers. We have provided a point-by-point response explaining how we have addressed each of the editors or reviewers’ comments. We look forward to the outcome of your assessment. Here is a point-by-point manner below.

We hope the revised version is now suitable for publication and look forward to hearing from you in due course.

Yours sincerely,

On behalf of the co-authors 

Biruk Bayleyegn

PONE-D-20-24807

Magnitude and associated factors of peripheral cytopenia among HIV-infected children attending at University of Gondar Specialized Referral Hospital, Northwest Ethiopia. PLOS ONE

Dear Dr. Bayleyegn,

Thank you for submitting your manuscript to PLOS ONE. After careful consideration, we feel that it has merit but does not fully meet PLOS ONE’s publication criteria as it currently stands. Therefore, we invite you to submit a revised version of the manuscript that addresses the points raised during the review process.

Please ensure you address each of the methodological concerns raised by the reviewers, and provide the requested details to ensure that your study is fully reproducible.

If applicable, we recommend that you deposit your laboratory protocols in protocols.io to enhance the reproducibility of your results. Protocols.io assigns your protocol its own identifier (DOI) so that it can be cited independently in the future. For instructions

see: http://journals.plos.org/plosone/s/submission-guidelines#loc-laboratory-protocols

We look forward to receiving your revised manuscript.

Authors response: Thanks for your great appreciation and offer.

Journal Requirements:

Authors response: Thank you for raising this point. We agree with your constructive comment and as per your suggestion we re-organized and rewriting in accordance with the journals style throughout the revised version of the manuscript. 

Authors response: Thank you for your great looking. We provided the questioners as additional information in both local language and English in the revised manuscript labeled as “S1”. 

3. You indicated that you had ethical approval for your study. In your Methods section, please ensure you have also stated whether you obtained consent from parents or guardians of the minors included in the study or whether the research ethics committee or IRB specifically waived the need for their consent.

Authors response: Thank you for your suggestion. It would have been interesting to explore this aspect in the methodological parts of the revised manuscript and it was done as per your comments. This is stated in the methods section of the revised manuscript (at the top of page 6).

4. In the Methods section, please provide the specific sequences of the PCR primers used in your study.

Authors response: You well come. This is an interesting comment. In this study we use specific primers such as “GAG specific to HIV1 primer” in amplification of HIV1 RNA and it was well described in the “methods section, HIV RNA detection and quantification subsection” of the revised manuscript (page 8). 

"First, the authors would like express their deepest gratitude to Department of Hematology and Immunohematology, School of Biomedical and Laboratory Sciences, College of Medicine and Health Sciences, University of Gondar for financial support and allowing them to conduct this research."

"The authors received no specific funding for this work."

Authors response: Agreed. This was our typesetting mistake. The University of Gondar only give the chance to do this research rather giving financial support. We remove funding related statement from the revised manuscript and no need of updated funding statement. We correct it in the acknowledgment part of the revised manuscript (page 28) 

6. We noticed you have some minor occurrence of overlapping text with the following previous publication(s), which needs to be addressed:

https://www.omicsonline.org/scientific-reports/JIDT-SR-662.pdf

https://link.springer.com/chapter/10.1007%2F978-3-030-35433-6_10

http://medcraveonline.com/HTIJ/gender-based-differences-in-hematological-and-cd4-t-lymphocyte-counts-among-hiv-patients-in-ido-ekiti.html

In your revision ensure you cite all your sources (including your own works), and quote or rephrase any duplicated text outside the methods section. Further consideration is dependent on these concerns being addressed.

Authors response: Thank you for addressing your constructive comments. Most of the above published articles were used in our paper and cited properly in order to give credit to each article (Reference 30 and Reference 9). This idea taken from these literatures were also rephrase in accordance with our paper and please be confident that no overlapping of the text with any papers. 

Comments to the Author

1. Is the manuscript technically sound, and do the data support the conclusions?

Reviewer #1: Partly

Reviewer #2: Yes

Authors response: Thank you for your appreciation. This research is done in collaboration with the most senior scientists working at the University of Gondar and by incorporating their comments and suggestion before submitting to the journal. So, we feel that this paper is soundful. ________________________________________

2. Has the statistical analysis been performed appropriately and rigorously?

Reviewer #1: No

Reviewer #2: No

Authors response: Thank you. The statistical part of the manuscript was well performed by statisticians who had well experienced in the field. In this study, the statistician only done statistical analysis for the major cell lineage (anemia and leukopenia) rather neutropenia and lymphopenia in order to limit number of tables. However, we didn’t do statical analysis of one of the three cell lineages “Thrombocytopenia” with independent factors due to that thrombocytopenia was detected among 4 children which is not fulfill the binary logistic regression criteria to predict the association between an exposer and the outcome. ________________________________________3. Have the authors made all data underlying the findings in their manuscript fully available?

Reviewer #1: Yes

Reviewer #2: Yes

Authors response: Thank you for your encouragement. We also added the questioners used in this stud as a supplementary information using local language and English version in the revised manuscript. 

4. Is the manuscript presented in an intelligible fashion and written in standard English?

Reviewer #1: No

Reviewer #2: Yes

Authors response: Thank you. We corrected typographical and grammatical errors throughout the revised version of the manuscript.

5. Review Comments to the Author

Reviewer #1: 

• Thank you for inviting me to review a manuscript entitled with “Magnitude and associated factors of peripheral cytopenia among HIV-infected children attending at University of Gondar Specialized Referral Hospital, Northwest Ethiopia”

• Why none probable convenient sampling technique was used to select the study participant? b/c the result will not be representative

Authors response: Thank you again. In the present study we applied none probable convenient sampling technique to recruited 255 HIV infected children. This was because of that it was impossible to draw random probability sampling techniques due to time and cost considerations. Moreover, as the study also included new HIV infected children it is impossible to use other technique to recruited this child as it is rare case now adays. By this reason we prefer the none probable convenient technique.

• The sample size is small since the study was conducted solely in a single institution. In addition, why not you didn’t take previous proportions which maximize your sample size. Please justify why? 

Authors response: We agree with you and thanks for your critical insight. As it was conducted among HIV infected children, there were only 536 HIV infected children during the study period in the Hospital ART registration logbook which was less than 10,000. Then by computing finite population correction formula the total sample size was 203. However, in order to maximize the sample size, we included 255 HIV infected children from 536 which is above the required sample size. In addition, this study was only conducted in a single institution and stated as one of the study limitations, look the limitation part of the revised manuscript (bottom of page 26) 

• Why not you didn’t used a case control study to assess factors?

• Authors response: This is an interesting question. Thank you, the reviewer. We agree with you. However, it is difficult to get equal proportion of HAART NAÏVE and HAART experienced children. Now a days new HIV infected child is rare due to Prevention of mother-to-child transmission (PMTCT) programs which offers a range of services for women of reproductive age living with or at risk of HIV to maintain their health and stop their infants from acquiring HIV. Furthermore, in this study only included children between 6 months to 15 years. But according to WHO recommendation the children began HAART immediately after confirmed HIV positive which is impossible to get HAART NAÏVE children who were less than six months. By this reason it is difficult to get HAART naïve children to compare with HAART experienced children in order to assess the independent factors. By this reason we apply cross sectional study design in order to include all HAART naïve and HARRT experienced children. 

• Ethical clearance: How blood specimens were collected while most children with HIV are anemic, other hematologic complication and malnutrition? Please send the scan of the IRB permission letter

Authors response: This is an interesting comment. Give you thanks. All study participants who were severely ill were excluded in the study and didn’t give any sample. After explain the purpose and risk of the procedures all voluntary study participants were give their blood sample based on Ethical approval obtained from the ethical and review committee of University of Gondar attached here with.

• How did you get each participant since thy may come at different period of follow up period

Authors response: Thank you for pointing out this. Based on the new WHO recommendation each child have appointing once per month in order to take their ART treatment on the ART clinic and in order to check their health status every month. So, during the 3 months of the study period, all the data collectors and sample collectors were available on the ART clinic and we get each child until the required sample size was obtained. During this study period many more children were contact repeatedly and we only take samples once from each individual.

• Why you select viral load as a main covariate than other independent variables.

Authors response: Good looking. Thank you for your constructive comments. HIV associated hematological abnormalities seem to be dependent on the level of viral replication. These abnormalities are severing among study participants in the late stages of HIV at high viremia and decreased CD4 count. Thus, viral load testing has been viewed as the best predictor of the risk of developing AIDS-related complications and can be used to monitor disease progression; determine prognosis; select patients for therapeutic trials and monitor therapy. This is due to that children are potentially at a higher risk of developing viremia due to weight-based dosing, poor tolerability of drugs and suboptimal adherence. However, CD4+ T cell counts alone seems to be an inadequate immunological parameter to measure prognosis and anti-retroviral therapy. Moreover, in recent WHO’s recommendations, plasma (RNA) viral load is a good marker of therapeutic adherence, disease progression and treatment efficacy as well as the main therapeutic follow-up parameter rather than CD4 counts. By considering this point in this study, viral load determination was the main independent factors of cytopenia among HIV infected children and it was stated shortly at the bottom of the 1st paragraph of the introduction part.

• It is not recommended to use the term predictors in cross-sectional study please change it with associated factors throughout the paper.

Authors response: Thank you for your critical suggestions. We mad it associated factors/independent factors instead of predictors throughout the revised manuscript.

• You run a logistic regression for only 2 outcome variables (anemia and leukopenia) why not you assessed factors of other components of cytopenia?

Authors response: Great looking of this paper. We agree with your constructive comments. However, in this study Thrombocytopenia was detected among 4 children which is unable to perform any association of the dependent and independent variables. In this study we didn’t do statical analysis of one of the three cell lineages “Thrombocytopenia” with independent factors due to that thrombocytopenia was detected among 4 children which is not fulfill the binary logistic regression criteria to predict the association between an exposer and the outcome.

• Why not you used other logistic regression model when you have more than 1 outcome variable

Authors response: Thank you the reviewer for your nice looking. In this study we use binary logistic regression to determine the descriptive value and multi-variable logistic regression to assess the association of dependent and independent variables which is more preferable than other confusing models.

• The discussion is not in detail, not including other studies done in Ethiopia among HIV positive children.

Authors response: Thank you for your suggestion. As much as possible we included the international and local studies to compare the finding of the current study. Notably, in Ethiopia there were no any more studies illustrated about cytopenia among HIV infected children. We have seen puplished studies for appropriateness of our study and most of the studies were reported about only Cytopenia on HAART, or only cytopenia among HAART naïve. But the current study included both HAART Naïve and HAART experienced children which is difficult to compare the current finding with the previous study. So, by considering this point we only included the studies conducted in Ethiopia which stats about magnitude of cytopenia among HIV infected children whether they were HAART naïve or HAART experienced in order to compare with the current finding. 

Reviewer #2: 

Comments to authors

I am happy by reviewing this manuscript. For improving the quality of the paper, the following comments were forwarded for the authors.

Title: Magnitude and associated factors of peripheral cytopenia among HIV infected children at the University of Gondar Specialized Hospital ART clinic, Northwest Ethiopia.

1. In the abstract, Background section, the author didn’t show the need/argument in the study. In the result section, the factors which affect cytopenia is not stated rather the author depicts merely for anemia. Besides, the conclusion is not in line with the title.

Authors response: Thank you the reviewer for your critical comments. We were agreed with you and corrected as per your comments. In the abstract section we stated the main objective of the study separately from the background section and explained the need of the study. In the result part of this study, the analysis of the association was performed separately with dependent variables such as anemia and leukopenia instead of cytopenia as per the definition of cytopenia is the deficiency of either of the three cell liang in the blood (anemia, leukopenia, thrombocytopenia). So, the major independent factors were associated with anemia rather leukopenia. 

2. In the Methods and materials part, why the author didn’t include the children less than 6 months. it is a critical question

Authors response: This is an interesting comment. Thank you again the reviewer. We agree with you. However, children less than 6 month were excluded in the study. This is because of that duration of HAART was one of the independent factors and hence this child taken HARRT less than 6 months were not appropriate to predict whether it is an associated factor or not. In order to conclude duration of HAART is an associated factor of cytopenia the study participants should take treatment not less than 6 months. By this reason children less than six month were not included in this study. 

3. In the result part, please re write the way of putting Interquartile range (IQR) with median 

Authors response: Thank you. We corrected it in the result section of the revised manuscript.

4. How can you operationalize the presence or absence of intestinal parasite for this particular study?

Authors response: Thank you for raising this point. It is clear that the presence or absence of intestinal parasite means that whether an individual harbor one or more trophozoite/cyst, larvae, or ova of intestinal parasites or not during stool examination. This is stated in the method section, parasitological examination part (bottom of page 8).

5. Be consistent while you write “multivariable versus multivariate analysis”

Authors response: Thank you again. We agree with you and we corrected it throughout the revised manuscript by using multi-variable instead of multivariate.

6. In the result section, you didn’t touch your title because the author operationalized Peripheral cytopenia as a reduction in either of the cellular elements of blood. The author solely done multivariable model for each parameter independently (anemia, and leucopenia)

Authors response: You have raising an interesting point here. As you stated peripheral cytopenia is defined as reduction in either of the cellular elements of blood, i.e., RBCs, WBCs or platelets. Based on this definition if an individual had one cell reduction, he/she was cytopenic and the analysis should perform separately anemia/leukopenia/thrombocytopenia verses independent factors. However, if one study participant had pancytopenia which is defined as reduction of the whole cell lineages, must perform multi-variable analysis of pancytopenia with independent variables. But in this study no pancytopenia was detected, whereas only high magnitude of uni-cytopenia and 10 bi-cytopenic cases were detected. So, based on this multi-variable logistic regression was applied separately such as anemia and leukopenia with independent factors and this is in line with the title. 

7. Besides, the author didn’t address the factors which predicts thrombocytopenia

Authors response: Thank you for your critical view. We didn’t do statical analysis of one of the three cell lineages “Thrombocytopenia” with independent factors due to that thrombocytopenia was detected among 4 children which is not fulfill the binary logistic regression criteria to predict the association between an exposer and the outcome.

8. In the discussion part, remove unnecessary sentences like “In this study a total of 255 consecutive HIV infected children were evaluated for peripheral cytopenia on their follow up visit at UOGSRH ART clinic.”

Authors response: Agreed. These like errors were corrected throughout the revised manuscript. 

9. Correct all the above comments point by point

Authors response: Thank you all for your constructive comments once again. We have provided a point-by-point response explaining how we have addressed each of the editors or reviewers’ comments above.________________________________________

6. PLOS authors have the option to publish the peer review history of their article (what does this mean?). If published, this will include your full peer review and any attached files.

Do you want your identity to be public for this peer review? For information about this choice, including consent withdrawal, please see our Privacy Policy.

Reviewer #1: Yes: Biruk Beletew Abate

Reviewer #2: Yes: Mohammed Ahmed

While revising your submission, please upload your figure files to the Preflight Analysis and Conversion Engine (PACE) digital diagnostic tool, 

https://pacev2.apexcovantage.com/. PACE helps ensure that figures meet PLOS requirements. To use PACE, you must first register as a user. Registration is free. Then, login and navigate to the UPLOAD tab, where you will find detailed instructions on how to use the tool. If you encounter any issues or have any questions when using PACE, please email PLOS at figures@plos.org. Please note that Supporting Information files do not need this step.

---

## [Editor Report · Decision Letter 1]

16 Feb 2021

Magnitude and associated factors of peripheral cytopenia among HIV-infected children attending at University of Gondar Specialized Referral Hospital, Northwest Ethiopia.

PONE-D-20-24807R1

Dear Dr. Bayleyegn,

We’re pleased to inform you that your manuscript has been judged scientifically suitable for publication and will be formally accepted for publication once it meets all outstanding technical requirements.

Kind regards,

Grzegorz Woźniakowski, Full professor, PhD, ScD

Academic Editor

PLOS ONE
---

## [Editor Report · Acceptance letter]

19 Feb 2021

PONE-D-20-24807R1 

Magnitude and associated factors of peripheral cytopenia among HIV-infected children attending at University of Gondar Specialized Referral Hospital, Northwest Ethiopia. 

Dear Dr. Bayleyegn:

I'm pleased to inform you that your manuscript has been deemed suitable for publication in PLOS ONE. Congratulations! Your manuscript is now with our production department. 

Kind regards, 

on behalf of

Prof. Grzegorz Woźniakowski 

Academic Editor

PLOS ONE